# Identifying Factors That Predict Behavioral Intention to Stay under Lockdown during the SARS-CoV-2 Pandemic Using a Structural Equation Model

**DOI:** 10.3390/ijerph19052757

**Published:** 2022-02-27

**Authors:** Joaquin Alberto Padilla-Bautista, Gilberto Manuel Galindo-Aldana

**Affiliations:** 1Research Group of Mental Health, Profession and Society, Laboratory of Psychosocial Research, Guadalupe Victoria Engineering and Business Faculty, Autonomous University of Baja California, Mexicali 21720, Mexico; 2Research Group of Mental Health, Profession and Society, Laboratory of Neurosciences and Cognition, Guadalupe Victoria Engineering and Business Faculty, Autonomous University of Baja California, Mexicali 21720, Mexico; gilberto.galindo.aldana@uabc.edu.mx

**Keywords:** quarantine, lockdown, isolation, pandemic, attitude, social distancing

## Abstract

Lockdown is considered to be a successful strategy for preventing the spread of severe acute respiratory syndrome coronavirus 2 (SARS-CoV-2). To predict the *behavioral intention* to stay under lockdown (BIKL), components of the theory of planned behavior (TPB) and the behavioral indicators of infection were applied. Sampling was conducted between 11 April and 30 May 2020. The objective of the study was to identify factors predictive of BIKL by means of a structural equation model. Method: A correlational and comparative repeated measures study was conducted with a sample of 315 participants from different cities in Mexico. Results: Model indices were *χ*^2^ = 505.1, *SD* = 228, *p* < 0.001, *χ*^2^/*SD* = 2.2, CFI = 0.91, RMSEA = 0.06, and SRMR = 0.06; 47% of BIKL was explained by the variables *attitude* (*β* = 0.71, *p* < 0.001), *subjective norm* (*β* = 0.14, *p* = 0.042), and *behavioral control* (*β* = 0.24, *p* < 0.001). Conclusions: Personal and family work conviction and persuasion are favorable for the maintenance of lockdown, including concepts of civic responsibility, a positive attitude, and a family that supports compliance with lockdown. From a governmental point of view, there is a context that promotes control over the situation and exerts a positive impact on the behavioral intention to stay under lockdown.

## 1. Introduction

On 31 December 2019, the World Health Organization (WHO) [1] was informed of approximately 27 cases of pneumonia with an unknown etiology; the causative agent was identified as a new virus in the Coronaviridae family, later named coronavirus 2, which causes severe acute respiratory syndrome (SARS-CoV-2). The clinical syndrome of this virus was named coronavirus disease 2019 (COVID-19). Afterward, SARS-CoV-2 spread around the world, and case counts increased exponentially, so on 11 March 2020, the WHO declared a global pandemic due to COVID-19, the first pandemic generated by a coronavirus. As there was no vaccine available to control this pandemic, the WHO recommended that countries adopt other strategies that were reliant on the participation of the government and the entire society. Each country had to generate a comprehensive strategy to prevent the spread of infection, save lives, and minimize the effects of the pandemic. Simultaneously, the WHO emphasized that there must be a delicate balance between health protection, the reduction of social disorganization, and respect for human rights. An effective strategy for dealing with a pandemic is to prevent the spread of disease by reducing the mobility of citizens and by separating healthy people from people who may be sick or infected with the virus [2,3]. This is achieved through several measures: quarantine, which allows separating and restricting people who have been exposed to a disease or who have traveled to an affected region and may not be infected or may be asymptomatic [4]; isolation, which protects uninfected people from people with a confirmed diagnosis of contagious disease [5]; and social distancing, which is implemented to reduce physical interaction and contact between persons [6] and is complemented by the use of masks, maintaining a distance of at least six feet between persons, avoiding crowds in public or private places, and working from home whenever possible. Finally, lockdown, which aims to restrict the movement of people, can be imposed when there is a large number of people who have become ill within an entire region or country. In these cases, the government can order the closure of public spaces and transportation while maintaining essential activities such as health services.

Mandatory lockdown may not be an option for specific regions because of characteristics such as access to goods, economy or culture. For example, in some countries in the Americas, with people living in social and economic inequality and where 43.9% of the population lives below the poverty line, it is impossible to apply an absolute lockdown as a mandatory measure since the population lacks the economic saving capacity and the necessary resources to face extreme mobility-limiting measures [7]. It is well known that the measures of confinement often worsen the conditions of the most unprotected people since they cause people to become unemployed and evicted from their homes [8]. Lockdown is more difficult for those who live in small houses, in houses without gardens, or share their living spaces and bathrooms [9]. The problems of privacy in families cohabitating in small spaces are amplified when intimacy is required in a couple’s relationship and there is no private bedroom. This is exasperated by the fact that many homes in Mexico where nuclear families (composed of a married couple and children) live have house-extended family members such as grandparents, uncles, aunts, and nephews [10]. Sociocultural differences can be exacerbated when a combination of political, social, and economic causes limits access to legal rights and welfare services [1]. In addition to economic problems, some people face other issues such as abuse, human rights violations, impunity, slow procurement, and justice administration, which are accompanied by a high incidence of violence.

Measures to mitigate and control COVID-19 and, along with it, social mobility, such as the interruption of nonessential activities, have been aimed at the means of living and not directly at the people. These measures have been implemented by means of convincing arguments and persuasion, but not by obligation or force [7,11]. In addition, limiting mobility in a population with low resources is essential since the use of public spaces such as public transportation favors contagion due to lack of ventilation, overcrowding, and a prolonged stay in confined spaces [12]. Situations in some countries highlight how pandemics often affect the weakest populations the most in emergency situations [13]. These situations clarify why the government has implemented a lockdown in which citizens are responsible for reducing contagion; from a psychosocial point of view, this approach addresses the importance of individuals as autonomous beings who have the capacity to ensure the care of their own health [14]. In this way, individuals consider the probable consequences of their behavior as based on normal expectations. However, there is no clear indicator that can identify when people have applied a voluntary intention to their actions or when they [15] have stopped thinking about the decisions that they will make in each situation that is presented to them [16].

For this research, the theory of planned behavior (TPB) model was implemented to explore the variables predictive of behavioral intention to remain under lockdown (BIKL), as this theory has provided important answers to health-disease problems [17]. According to the literature, the components of TPB are *attitude, behavioral intention, subjective norm*, and *perceived behavioral control* [18], and they are defined as follows: *behavioral*
*intention* is considered to be the immediate antecedent of behavior and refers to the self-estimation that a person makes about the probability of performing a certain behavior [19], and *attitude* is a latent disposition to respond favorably or unfavorably to a psychological objective that is developed through the beliefs that a person holds towards an object and the reason why the behavior of the individual is an object of evaluation.

In TPB, there are three types beliefs that one may acquire: (a) descriptive beliefs, which are developed by direct observation, (b) inferential beliefs, which are obtained through contact with another person or by characteristics observed from an audience, and (c) beliefs, which are expressed by others about an object. The *subjective norm* is the set of beliefs that the individual thinks that people have that are significant to the individual and their perception of what others expect them to do. Finally, perceived *behavioral control* is a set of beliefs that the individual has about whether he or she has the necessary capacity or ability to perform a certain behavior. Another factor is whether the environment favors or hinders the behavior. In addition to the beliefs exposed according to TPB, there is a question of whether the beliefs that the individual has about the probability of contagion are also determining factors in the prediction of *behavioral intention*. Specifically, the hypothesis of this project states that a perception of greater contagion in the city (vulnerability) and less contagion at home (security) will also affect BIKL, and it is also believed that the number of infected and deceased individuals has a positive effect on maintaining *behavioral intention*. Considering this background, the goal of this research was to identify determining factors that predict the *behavioral intention* to remain under lockdown by using a structural equation model.

## 2. Materials and Methods

### 2.1. Research Design

For the abovementioned objective, we applied a descriptive nonexperimental correlational and related sample comparative design.

### 2.2. Description of the Sample

Participants were recruited by an open invitation to participate via social media.

The inclusion criteria included Mexican citizens older than 18 years old, participants who gave informed consent, and those with complete data. During the first assessment, sociodemographic information, such as age, sex, and TPB variables related to contagion beliefs at home and in the city, as well as the variables related to awareness about rates of infected and deceased people, was gathered. After the second confinement period measure (phase one), the same metrics were applied as in phase one; however, the *behavioral intention* measure was changed since the first measure did not produce adequate validity and reliability indices.

### 2.3. Instruments

To measure possible predictors of BIKL, we included positive *attitude* factors, *subjective* support norms, *behavioral resource control*, and *behavioral intention*, all from the instrument to measure *attachment to lockdown*. The questionnaire was delivered in Spanish as the native language of the participants. Administered items, ordinal alpha coefficients, and response options are shown in Table 1 (please refer to Appendix A for a full version of the questionnaire). To measure *vulnerability*
*belief,* in the context of security, participants were asked the following questions: *how much infection risk do you think is in your city?* and, *how much infection risk do you think is in your home?* Options for response were as follows: 1 = No risk, 2 = Very low risk, 3 = Low risk, 4 = High risk, and 6 = Very high risk. To determine the *number of infections belief* and the *number of deceased*, the following questions were asked: *how many COVID-19-infected persons do you actually know?* and, *how many COVID-19 deceased persons do you actually know?* Responses to these questions were gathered using numerical codes.

### 2.4. Procedure

The instrument was delivered in two stages, with the second stage divided into two further phases: the first stage was from 11 to 26 April 2020, the second phase of confinement lasted from 24 March until 21 April, and the third phase was from 21 April to 30 May 2020.

The second assessment was 46 days after the first one, which was from 27 May to 4 June 2020, and occurred during the third phase of confinement in Mexico and the beginning of the activity reincorporation phase.

The questionnaire was administered via a Google Docs Form and was distributed by means of social media (e.g., Facebook, WhatsApp), explaining that it was research carried out by the Autonomous University of Baja California. The questionnaire was open to everyone, and those who answered the forms participated voluntarily. The average response time was 15 min.

### 2.5. Data Analysis

Reliability was measured using ordinal alpha [20] and was calculated from the average of the polychoric correlation matrix obtained through the Factor Analysis Program Software Package version 12.01.02 (Rovira I Virgili University, Tarragona, Spain).

We used Mann–Whitney’s *U* test to determine whether there were differences between attitude, *subjective norms*, *behavioral control*, and beliefs regarding the risk of contagion in the city and home and the perceived number of infected and deceased people based on whether participants responded to only the first or both questionnaires. To compare differences among attitudes, *subjective norms*, *behavioral control*, beliefs regarding the risk of contagion in the city and at home, and the perceived number of infected and deceased people at the time of the questionnaire, we conducted a repeated measures Student’s *t* test. For these and subsequent analyses, the sample included 315 participants.

To describe the magnitude and relevance of both analyses, the effect size was categorized as small (0.20), medium (0.50) or large (0.80), and the statistical power (1-*β*) was also calculated, with 0.80 being the minimum threshold to consider the results valid and not belonging to the category of a Type II error. Both estimators were calculated using G*Power.

Afterward, we looked for the predictive capacity of *attitude, subjective norm, behavioral control, belief of contagion* in the city and at home, behavioral intention, and the number of people infected and deceased by COVID-19 variables for BIKL. A structural equation model was constructed using AMOS 24, with *behavioral* intention as the dependent variable and all of the other variables as independent variables. A maximum likelihood analysis was implemented, which, unlike asymptotic free distribution methods, requires at least a sample of 200 participants for parameter estimation. Additional parameters indicating the goodness of fit of the model were the relative chi-square (*χ*^2^/DF) ≤ 2, adjusted comparative fit index (CFI) ≥ 0.90, root mean square error of approximation (RMSEA) ≤ 0.08, standardized root mean square residual (SRMR) ≤ 0.10, and Tucker–Lewis coefficient (TLI). The adjustment indices for parsimony indicating an acceptable adjustment were PNFI and PCFI ≥ 0.60 [21]. For this analysis, only the data from the second assessment were used.

### 2.6. Ethical Considerations

All participants provided their informed consent on the application form before beginning to respond and claimed to be over 18 years of age at the time of application. The protection of the data was carried out in accordance with Federal Law on the protection of personal data held by individuals [22]. The method and procedures were ethically approved by the Ethics and Research Committee of the Autonomous University of Baja California (registration number POSG-020-1-05).

## 3. Results

A total of 1003 participants were recruited, which included 817 (81.5%) females and 186 (18.5%) males who were aged between 18 and 83 years old, with a median age of 33 years old, (25th percentile = 26.5, 50th percentile = 33.75, 75th percentile = 41, interquartile range = 12, *SD* = 11.34) 409 (40.8%) from in Mexico City, 156 (15.6%) from Baja California, 110 (11%) from Mexico State, 57 (5.3%) from Puebla, and 271 (27%) from different cities across Mexico. A total of 680 participants from the initial sample of 1003 agreed to receive the second study assessment by providing an e-mail address to receive the questionnaire individually. The final sample exhibited similar demographic proportions and was composed of 315 participants ranging in age from 18 to 66 years old with a median age of 32 years old, (25th percentile = 27.0, 50th percentile = 32.0, 75th percentile = 39, interquartile range = 12, *SD* = 10.3). Of these participants, 263 (83.5%) were female, 52 (16.5%) were male, 134 (42.5%) lived in Mexico City, 47 (14.9%) lived in Baja California, 37 (11.7%) lived in Mexico State, 14 (4.4%) lived in Puebla, and 83 (26.5%) lived in different cities in Mexico.

Ordinal alpha from both assessments showed 0.71 to 0.93 reliability coefficients that are considered appropriate for at least 10 items per factor model [23]. The ordinal alpha for the group of three dimensions from the first assessment was 0.93, and it was 0.96 for the group of four dimensions of the second assessment. The differences between participants who responded only to the first assessment and those who responded to both are shown in Table 2. For this analysis, we used a sample of 1003 participants who completed the first assessment, dividing the sample between those who had responded only to the first assessment and those who had responded to both.

Participants who answered both assessments had higher scores in attitude and *behavioral control*. However, the effect size was small, and the statistical power did not reach 0.80, which would allow us to conclude that the results are representative of the general population. To determine whether the time during the lockdown had an effect on the study variables, a repeated measures Student’s t test was conducted, and the results are shown in Table 3.

These results indicate that there was a difference for all variables between the two assessments; higher scores were observed with the second assessment for all variables except for *subjective norm*, which did not show statistically significant differences. Likewise, the effect sizes indicate that the magnitude of these differences ranged from low to medium. The statistical power was adequate for all comparisons.

When creating a model for intentional behavior to remain under lockdown, the first model employed all variables of the instrument and showed the following indices *χ*^2^ = 505.1, *SD* = 228, *p* < 0.001; *χ*^2^/*SD* = 2.2, CFI = 0.91, RMSEA = 0.06, SRMR = 0.06, PNFI = 0.70, PCFI = 0.75, and TLI = 0.89. This model explained 47% of the variance in *behavioral intention*; however, only the variables for TPB were predictive of outcome according to regression analysis: *attitude* (*β* = 0.71, *p* < 0.001), *subjective norm* (*β* = 0.14, *p* < 0.042), *behavioral control* (*β* = 0.24, *p* < 0.001), infections in the city (*β* = 0.001, *p* = 0.998), at-home infections (*β* = 0.001, *p* = 0.984), infected persons (*β* = −0.01, *p* = 0.190), and deceased persons (*β* = 0.005, *p* = 0.623).

In the second model, we included only those variables that were identified as possible predictors. In this analysis, the observed indices were as follows: *χ*^2^ = 424.71, *SD* = 164, *p* < 0.001; *χ*^2^/*SD* = 2.5, CFI = 0.91, RMSEA = 0.07, SRMR = 0.06, PNFI = 0.74, PCFI = 0.78, and TLI = 0.89, indicating appropriate model fit [24,25,26] and explaining 47% of the variance in *behavioral intention* with the following parameter estimates for variables in the model: *attitude* (*β* = 0.70, *p* < 0.001), *subjective norm* (*β* = 0.14, *p* = 0.045), and *behavioral*
*control* (*β* = 0.25, *p* = 0.001). A standardized image summarizing the results from this model is shown in Figure 1.

## 4. Discussion

Longitudinal comparison tests indicated that the mean scores of the components of TPB were high in both assessments, exceeding the theoretical mean for all factors. Despite this, *attitude* exhibited a significant increase, suggesting that the more time there is under lockdown, the greater the agreement of complying with it. This result may possibly be derived from observing the advantages or benefits that the individual or someone else obtained from the appropriate compliance with the lockdown. The immediate benefit observed was an impact on the participants’ finances, as 25% of participants reported a negative impact (“bad because I lost my job”). These data are consistent with those found by Salas [27], who reported that mitigation measures had an impact on employment levels, which translated into lower income for workers since the total or partial closure of economic activities translated into a lower number of hours worked.

Nevertheless, the item regarding the economic situation showed the following results: “the impact in your economy has been… (1) good, because it helped me to save money (59.5%), (2) bad, because I lost my job (12.3%), (3) none of these (7.6%), and (4) other (7.9%)”.

Regarding economic issues that may influence BIKL, Salas [27] found that although substantial increases in income inequality have been reported, the data reported only included income derived from work, without considering other forms of income such as social support, loans, employer support, family support or other sources of income such as the creation of small or medium-sized businesses created during the pandemic. This variety of sources of income and the savings in transportation costs due to the lack of mobility could explain why the pandemic was perceived as an opportunity to save money. This is consistent with other research that reported a trend in shifting spending priorities tending toward rational consumption and a focus on savings.

On the other hand, *subjective*
*norms* did not show a significant change but exhibited high scores in both assessments. This is consistent with the literature, which reported that the Mexican population is characterized by being oriented toward collectivism, with a culture whose identity is built from the social network to which it belongs, including *trust in others*, an interest in the *well-being of other people*, and a *dependency between individuals* [28,29].

An increase in perceived *behavioral control* suggests that a greater capacity and BIKL were acquired during the study period, and the entire context of the interruption of nonessential activities by the government facilitated the behavior.

The infection belief scores in the city were higher than the theoretical average, and the significant change matched the true case counts since in the 46 days between the assessments, 123,924 confirmed cases and 11,920 suspected cases were reported nationwide. Infection at home exhibited a significant increase, but both scores were below average, suggesting that home was perceived as a safe area during the pandemic. This is paradoxical in that keeping family close was associated with higher perceived risks of infection, yet collectivism encourages people to remain close to their social contacts [30]. The increase in the perception of infected people was consistent with true counts since the sum of confirmed and suspected cases was 135,844 cases nationwide.

The significant difference in the perception of deceased people may have been because, at the national level, there were 17,422 unfortunate deaths within the period between the assessments in this study. These data demonstrate that there was concurrent criterion validity with these variables [31].

All of the abovementioned significant differences showed effect sizes ranging from low to medium, and the statistical power exceeded the required minimum of 0.80 in all cases, which indicates that the 315 participants included in this study’s sample were sufficient to describe the effect of the lockdown on the population.

Although there were differences in all the independent variables, in the structural equation model, which had favorable goodness-of-fit metrics, only the variables that compose the TPB were shown to be predictors for behavioral intention to remain under lockdown. For this reason, it is necessary to work on campaigns that increase the positive *attitude* of individuals, where families encourage their members to remain committed to lockdown due to SARS-CoV-2, as well as working in a context that allows people to have the ability to control the situation. We must continue to promote conviction and persuasion in the face of a social influence in which the group’s attitudes and beliefs alter those of the individual, thus promoting conformity through acceptance; this allows people to believe that group norms are personal and sincere, so they internalize and demonstrate them publicly and privately [26]. These results are consistent with the finding that Chinese university students reported higher home maintenance behavior than American students, mainly because they belong to a collectivist culture similar to the Mexican culture [32]. These familiar circumstances may promote the belief that contagion not only affects them individually but also represents a risk factor for the group to which they belong [33]. On the other hand, social and individual behaviors are highly influenced by the environmental effects of different norms or regulations; according to other authors, a positive environment and attitudes can be encouraged and have an impact on the behavioral intention of populations [34].

Nonsignificant data are informative since they indicate that, at least in this population, highlighting the negative impact of the pandemic will not result in behavioral change since these variables were not found to be predictive in the model. Although lockdown is a strategy that helps prevent the spread of disease, a model that explains compliance with lockdown when the responsibility is placed on the individual is lacking. This is possibly because much of the population is aware that they are more likely to die of starvation than of COVID-19, and even before the pandemic, going to a hospital for treatment of a serious medical problem was not feasible because it was unaffordable. For many people, the risk of dying of high blood pressure or diabetes, rather than COVID-19, is real, as is their need to earn money and join the crowded lines at convenience stores to buy basic food for their family [9].

Although lockdown is a strategy that helps prevent the spread of diseases, there is a lack of a model to explain its maintenance when the responsibility for carrying it out lies with the individual. The population has a need for mobility because people rely on per diem income, they live in small spaces that make social distancing impossible due to family cohabitation, and they have a need to use public transportation. All of these elements are characteristic of a population that has a little private space and must make use of public spaces that facilitate contagion. Thus, this research contributes to our understanding of compliance with lockdowns by providing a model that predicts behavioral intention, which is considered the direct antecedent of behavior [19].

## 5. Limitations

One of the main limitations of the present study is that, even though the sample was large and representative of the population, the ability of the data to represent different social, cultural or political conditions that may help us to better understand the approach and behavior of different groups during the lockdown is limited. Finally, it is necessary to look for a safe way to obtain data from the population that does not have access to the internet since according to the National Survey on Availability and Use of Information Technologies in Households (ENDUTIH) National Institute of Statistics and Geography (INEGI) [35], only 70.1% of people in Mexico have internet access, so data from the most vulnerable people may not be included in the study. Further sampling is needed to determine the possible representativeness of the findings of this study.

Based on the results of this study, the hypothesis that belief of infection in the city and at home and knowledge of people infected and deceased are predictive of behavioral intention is not supported. Although significant differences were found when comparing participants who only completed one of the two assessments, these differences, as indicated by the effect size, were not very relevant. Likewise, the statistical power indicated that the findings are not generalizable to the entire population. Therefore, it cannot be inferred that the participants who responded to both assessments are different from those who completed only one of the assessments or that they have particular characteristics that could induce bias in this research.

The population used in this research is biased because it was an online application, so it is very likely that the survey was answered by people with internet access and leisure time, so it is necessary to analyze different vulnerable groups, such as homeless people, people in transit, and irregular, informal populations, migrants, refugees, with unstable residence or without identity documents; these populations are especially vulnerable because they have less access to vaccination campaigns [36] and are sharing a reduced public space with nationals and even with foreigners, making it difficult for them to become isolated and socially distanced [37].

## 6. Conclusions

Future research should include different populations and make use of different assessment forms. In our study, there was a single model for both sexes based on data from the sample of 315 people. It is of interest to design a study with a larger population and generate a model for each of the sexes separately. It would also be interesting to study lockdown in different contexts, specifically in those with the use of force to maintain lockdown.

## Figures and Tables

**Figure 1 ijerph-19-02757-f001:**
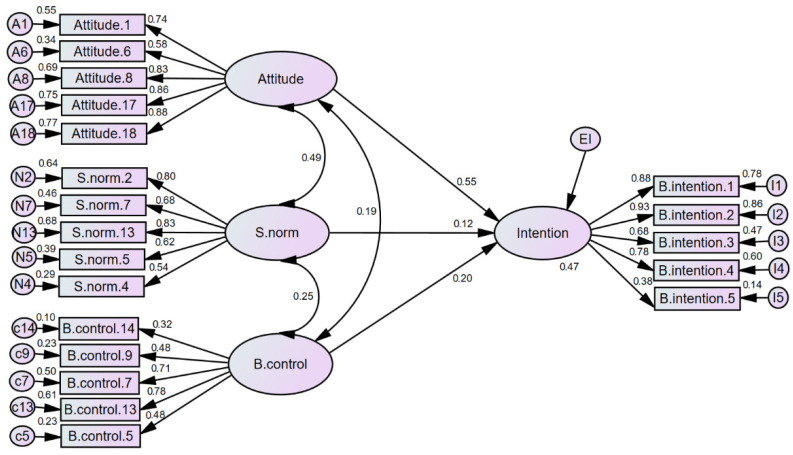
Predictive Model of *Behavioral Intention* to Remain Under Lockdown.

**Table 1 ijerph-19-02757-t001:** TPB items, response options, and ordinal alpha coefficients.

		Ordinal Alpha	
		First Measure	Second Measure
	**Attitude**	*n* = 1003	*n* = 315
Response options: **1 = not at all to 5 = a lot**	To keep myself under lockdown is:	0.86	0.93
	1. Necessary		
	2. Responsible		
	3. Intelligent		
	4. Good		
	5. Important		
	**Subjective norm**		
	My beloved ones (family, friends or partner) in this lockdown …	0.90	0.88
	6. think that it is right that I stay at home		
	7. think that the less I go out the less I will be at risk of infection		
	8. say it is important for the security of all		
	9. ask me to not go out		
	10. worry about my health		
	**Behavioral control**		
Response options: **1 = I am not in agreement to 5 = I totally agree**	During this lockdown …	0.71	0.77
	11. I have economic support		
	12. I have a healthy body		
	13. I know how to take care of myself		
	14. I know I have the necessary skills to get ahead		
	15. I have savings to get ahead		
	**Behavioral intention**		
	While this lockdown lasts I …	--	0.90
	16. will try to NOT go out		
	17. will attempt to keep myself in isolation		
	18. will try to not receive visits		
	19. will try to do everything from home		
	20. will seek to order food at home		

**Table 2 ijerph-19-02757-t002:** Mann–Whitney’s U test results between participants from one and both assessments.

	Rank Average	Sum. Ranks	*U*	*z*	*p*	*dz*	1-*β*
*Attitude* 1 *	486.36	334,618.00	97,602	−2.55	0.011	0.16	0.41
*Attitude* 2 **	536.15	168,888.00					
*Subjective norm* 1 *	503.42	346,354.00	107,382	−0.23	0.816	0.01	0.82
*Subjective norm* 2 **	498.90	157,152.00					
*Behavioral control* 1 *	482.22	331,768.50	94,752	−3.20	0.001	0.23	0.51
*Behavioral control* 2 **	545.20	171,738.00					
*Contagion at the city* 1 *	496.37	341,504.50	104,488	−0.96	0.337	0.08	0.63
*Contagion at the city* 2 **	514.29	162,001.50					
*Contagion at home* 1 *	500.84	344,574.50	107,558	−0.19	0.846	0.01	0.85
*Contagion at home* 2 **	504.54	158,931.50					
*Infected persons* 1 *	505.23	347,599.00	106,137	−0.60	0.549	0.03	0.59
*Infected persons* 2 **	494.94	155,907.00					
*Deceased persons* 1 *	501.19	344,816.50	107,800	−0.98	0.325	0.06	0.49
*Deceased persons* 2 **	503.78	158,689.50					

Note: * *n* = 688 participants who only completed the first assessment, ** *n* = 315 participants who completed the first and second assessments, *U* Mann–Whitney’s value, *z* is the standard score, *p* represents significance, and *dz* represents Cohen’s effect size.

**Table 3 ijerph-19-02757-t003:** Paired sample *t* test results.

	Min	Max	M	*SD*	*t*	*r^P^*	*p*	*dz*	1-*β*
*Attitude* 1°	1	5	4.40	0.60	5.13	0.602	0.001	0.29	0.97
*Attitude* 2°	1	5	**4.56**	0.61					
*Subjective norm* 1°	1.33	5	4.29	0.80	0.16	0.676	0.867	0	0.86
*Subjective norm* 2°	1.67	5	4.29	0.73					
*Behavioral control* 1°	2	5	3.93	0.67	7.32	0.741	0.001	0.41	0.99
*Behavioral control* 2°	1.67	5	**4.13**	0.67					
*Behavioral intention* 1°	-	-	-	-	-	-	-	-	-
*Behavioral intention* 2°	1	5	4.30	0.70					
*Infection in city* 1°	1	6	4.69	0.98	7.09	0.474	0.001	0.39	0.99
*Infection in city* 2°	1	6	**5.08**	0.95					
*Infection in home* 1°	1	5	2.74	0.06	3.35	0.530	0.001	0.24	0.86
*Infection in home* 2°	1	6	**2.94**	0.07					
*Infected persons* 1°	0	4	0.46	0.84	10.56	0.251	0.001	0.59	1
*Infected persons* 2°	0	40	**2.78**	4.01					
*Deceased persons* 1°	0	1	0.009	0.10	6.25	−0.017	0.001	0.35	0.99
*Deceased persons* 2°	0	60	**1.31**	3.70					

Note: *n* = 315 for all variables, except for the first application of the *behavioral intention* where the measure was changed, bold means significant; 1° = first application, 2° = second application; 46 days between the assessments.

## Data Availability

Data 1: Data first and second measurement 315 persons Identifying Factors That Predict Behavioral Intention to Stay under Lockdown during the SARS-CoV-2 Pandemic Using a Structural Equation Model are available: https://www.researchgate.net/project/Factors-That-Predict-Behavioral-Intention-to-Stay-under-Lock-down-during-the-SARS-CoV-2-Pandemic (accessed on 15 February 2022).

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
