# Peer review of "Identifying Factors That Predict Behavioral Intention to Stay under Lockdown during the SARS-CoV-2 Pandemic Using a Structural Equation Model"

_ijerph, 2022, doi:10.3390/ijerph19052757_

Round 1

Reviewer 1 Report

The manuscript is substantially amended and it is acceptable for publishing.

Keywords are still written in different fonts.

Line 36, 2020 is in brackets is this wrong referencing or spelling error?

Research design is now more appropriately written.

In the manuscript, the authors use participants and respondents for the same group. Use either of these terms throughout the whole manuscript.

Explain the age variable equally for both samples.

There are still many errors in the references section but improving. References 20 and 22 need to be erased. When the authors are mentioning software, they should use syntax like this one: "Data analysis was performed by SPSS statistical software package version 22.0 (SPSS Inc., Chicago, IL, USA).".

Author Response

We would like to thank the reviewer for the clarifications and valuable suggestions provided. In this document we attach a reference with the list of modifications, adjustments and improvements that have been made to the manuscript based on these suggestions. As well as the manuscript with the mentioned modifications marked in blue color and tracking of changes.

Reviewer 2 Report

Thank you for reviewing this new version, which incorporates the suggested modifications. 
The article has benefited and is more robust than the previous version. 

Author Response

We would like to thank the reviewer for the valuable contributions and precise recommendations made to this manuscript.

Reviewer 3 Report

Despite the author that suggested implementation has been made in introduction section I don't see any suggested changes. Over the last period, considerable attention has been paid to the less well-off classes in the context of the pandemic. I am convinced that mentioning this in the work would improve the level and relevance of the article. I suggest to mention in relation to the introduction where reference is made to the classes most at risk and to the conclusions where an effective vaccination campaign is requested, two important works relating to the less affluent social status as a risk factor for covid relating to difficulties in conducting quarantine and isolation. (PMID: 34155485).

Recent literature is paying attention to the risk of getting sick with covid, the lower access to treatment, vaccination and the risk of more serious disease in the lower social classes (PMID: 35055541, and PMID: 33177039). It would be interesting if the author were able to connect the low-income populations with greater risk of disease and difficulties in conducting quarantine and social distancing.

Author Response

We would like to thank the reviewer for the clarifications and valuable suggestions provided. In this document we attach a reference with the list of modifications, adjustments and improvements that have been made to the manuscript based on these suggestions. As well as the manuscript with the mentioned modifications marked in blue color and trackingof changes.

This manuscript is a resubmission of an earlier submission. The following is a list of the peer review reports and author responses from that submission.

Round 1

Reviewer 1 Report

I read with great interest this manuscript by Padilla-Bautista et al. While the article is slightly different from what one could expect to see in a scientific paper (e.g. in terms of structure and weight given to the Introduction, Methods, and Discussion, at the expenses of the Results), the reading is enjoyable and the topic is of the utmost interest. It deals with a psychosocial and very important topic from a medical point of view. The study is really well done and reports interesting data in a clear and precise way. The bibliography takes into consideration works of excellent quality among the most prestigious journals, citing the most avant-garde works on the topic. I suggest stimulating the hottest topic that the author mentions in the manuscript, namely that of the less well-off classes as a risk factor for covid.

I also suggest to mention in relation to the introduction where reference is made to the classes most at risk and to the conclusions where an effective vaccination campaign is requested, two important works relating to the less affluent social status as a risk factor for covid relating to difficulties in conducting quarantine and isolation. (PMID: 34155485).

Recent literature is paying attention to the risk of getting sick with covid, the lower access to treatment, vaccination and the risk of more serious disease in the lower social classes (PMID: 32819364, or  PMID: 33177039). It would be interesting if the author were able to connect the low-income populations with greater risk of disease and difficulties in conducting quarantine and social distancing.

Author Response

We thank reviewer's comments and suggestions, the attached table presents the applied corrections.

Reviewer 2 Report

I thank the authors and the editor for giving me the opportunity to read the present work.

My main concerns indeed are 1) the author’s conceptualization of the relationship between their variables, 2) The introduction does not allow us to understand the research problem, 3) the instruments section should be described in more detail, 4) there are problems in the English language that make it difficult to understand the text, which is why it is difficult to assess the relevance of the analysis and the results obtained. 

Author Response

(The authors gave the same response as above.)

Reviewer 3 Report

The article title should be rephrased to exclude ambiguity regarding planned and predicted behavior.

The abstract should have the date when that study was done.

Numbers greater than 10 should be stated with numbers, not words.

The authors need to correct the decimals for each represented value, need to present the same number of decimals throughout the whole document.

The last two keywords are not in the MESH PubMed database and should be changed.

The first paragraph of the Introduction needs referencing.

The first 4 sentences are coupled and need to be omitted.

The authors need to check the instruction for authors and correct referencing style accordingly.

Line 35 to 39 is general knowledge and also not entirely correct and should be omitted or greatly improved.

Line 45-46 should be explained in more detail. Also, needs to explain the distinction between quarantine, isolation, lockdown, and curfew.

I believe that the expression quarantine is wrongly picked and should be changed with lockdown or curfew in the whole manuscript.

The authors should check the English language and style.

Abbreviations when first introduced should be explained, it is not enough to explain them in the Abstract and not in the manuscript.

The introduction should be focused on the main goal of this manuscript. Authors should not write about common knowledge and definitions related to the subject of their manuscript because they are addressing educated readers who are very well familiar with the current subject.

The goals stated in the Introduction section should be consistent with the objective stated in the Abstract.

Research design is in total wrongly written. Please consult relevant epidemiological literature dealing with the epidemiological methodology description.

Please explain recruitment for online survey participants and provide a cohort recruitment diagram within the Materials and methods section of your manuscript.

The median and interquartile ranges are more appropriate for presenting age data due to the fact that the distribution is probably not normal.

The authors should also elaborate on the representativeness of the sample for the study population.

The discussion section should be focused on the authors' own research results and their explanation comparison with similar studies.

The limitations stated within the conclusion section of the manuscript should be moved to the discussion section.

The reference list should be rewritten according to the instruction for authors.

The manuscript written in this way does not provide a sufficient amount of novelty and originality that is necessary for its publication.

Author Response

(The authors gave the same response as above.)

Round 2

Reviewer 2 Report

This version of the manuscript is much more robust and I am very happy to see that the authors engaged with the revisions so thoroughly.

I think the method can be improve, with the description of the measures deeply. 

Thank you again for letting me read your work and wishing you all the best in your endeavours.

Author Response

We thank once again the helpful and professional suggestions provided by the reviewer. We take this opportunity to submit a new version of the manuscript with the recommended corrections, as well as further improvements identified as necessary to increase the research report quality.

Reviewer 3 Report

The manuscript was greatly improved but there are still some things that need to be corrected.

There are still many grammar and English styling errors. I suggest that authors use additional services for English editing to fix many wording errors.

The authors use a different font type than instructed in the guidelines. Also, the formatting of the text (width of the paragraph) needs correction.

Referencing was again wrongly done. How can the first reference be number 2? How can abstract contain reference? References paragraph needs to be corrected according to the guidelines.

Research design is the same as before. It needs to be rewritten as mentioned the last time.

Lines 339-343 need rewording.

Limitations are only dealing with internet access and not with the sample size which is a more important problem, so it needs to be updated.

Author Response

(The authors gave the same response as above.)
